# Public Health Research Priorities for Fungal Diseases: A Multidisciplinary Approach to Save Lives

**DOI:** 10.3390/jof9080820

**Published:** 2023-08-03

**Authors:** Dallas J. Smith, Jeremy A. W. Gold, Kaitlin Benedict, Karen Wu, Meghan Lyman, Alexander Jordan, Narda Medina, Shawn R. Lockhart, D. Joseph Sexton, Nancy A. Chow, Brendan R. Jackson, Anastasia P. Litvintseva, Mitsuru Toda, Tom Chiller

**Affiliations:** Mycotic Diseases Branch, Division of Foodborne, Waterborne and Environmental Diseases, Centers for Disease Control and Prevention, Atlanta, GA 30329, USA; leo3@cdc.gov (J.A.W.G.); jsy8@cdc.gov (K.B.); vom5@cdc.gov (K.W.); yeo4@cdc.gov (M.L.); noq1@cdc.gov (A.J.); tup6@cdc.gov (N.M.); gyi2@cdc.gov (S.R.L.); ogi3@cdc.gov (D.J.S.); yln3@cdc.gov (N.A.C.); iyn0@cdc.gov (B.R.J.); frq8@cdc.gov (A.P.L.); nrk7@cdc.gov (M.T.)

**Keywords:** fungal diseases, research priorities, fungal disease surveillance, fungal disease diagnostic tests, fungal disease treatment

## Abstract

Fungal infections can cause severe disease and death and impose a substantial economic burden on healthcare systems. Public health research requires a multidisciplinary approach and is essential to help save lives and prevent disability from fungal diseases. In this manuscript, we outline the main public health research priorities for fungal diseases, including the measurement of the fungal disease burden and distribution and the need for improved diagnostics, therapeutics, and vaccines. Characterizing the public health, economic, health system, and individual burden caused by fungal diseases can provide critical insights to promote better prevention and treatment. The development and validation of fungal diagnostic tests that are rapid, accurate, and cost-effective can improve testing practices. Understanding best practices for antifungal prophylaxis can optimize prevention in at-risk populations, while research on antifungal resistance can improve patient outcomes. Investment in vaccines may eliminate certain fungal diseases or lower incidence and mortality. Public health research priorities and approaches may vary by fungal pathogen.

## 1. Introduction

Fungal diseases can range from mild skin infections to invasive diseases causing severe illness and death. These diseases also impose a substantial economic burden on patients and healthcare systems, with direct and indirect costs estimated at more than USD 11.5 billion each year in the United States [1]. Fungal disease health disparities place an uneven burden on certain populations, leading to increased morbidity and mortality [2,3,4,5,6]. 

Funding for fungal diseases is low in relation to their burden [7]. The global fungal research community is a small but highly motivated and diverse group of multidisciplinary researchers. With a united approach, fungal disease researchers can pool resources and knowledge to improve the understanding of fungal disease detection, prevention, diagnosis, and treatment. 

This manuscript aims to outline overarching and disease-specific public health research priorities for fungal diseases. The Centers for Disease Control and Prevention’s Mycotic Diseases Branch collaborated with public and private partners to prevent and control fungal diseases and identified these research priorities to serve as a roadmap for research entities, academics, and private groups to develop strategies to better understand and prevent fungal diseases. Furthermore, these priorities are meant to build and expand upon the World Health Organization’s fungal priority pathogens list [8]. The research needs outlined here are not comprehensive and will evolve with emerging data and studies. 

## 2. Priorities That Span Fungal Diseases (Table 1)

### 2.1. Burden and Distribution of Fungal Diseases

Characterizing the public health, economic, health system, and individual burden imposed by fungal diseases can yield valuable insights to promote and guide better prevention, treatment, and funding opportunities. Most fungal diseases are not reportable to public health authorities and have no local, national, or regional surveillance; thus, current fungal surveillance systems in the United States and elsewhere are limited by the geographic range from which data are collected and the depth of variables [9]. For example, coccidioidomycosis and *Candida auris* are nationally notifiable, while blastomycosis and histoplasmosis are only reportable in several states [9]. The underdiagnosis and misdiagnosis of fungal diseases also contribute to substantial gaps in understanding the disease burden. Establishing expanded national and regional surveillance to track the fungal disease distribution alongside early warning systems can identify emerging fungal disease threats. The National Notifiable Disease Surveillance System collects data from all states reporting certain fungal diseases, but the variables collected for these diseases are limited. Further exploration of the burden of fungal diseases associated with antifungal resistance or other conditions is warranted as they may present concurrently or predispose patients to both [4,10,11,12]. Besides surveillance, alternative methods to estimate the disease burden (e.g., number of cases, deaths, and costs) include analyzing healthcare administrative data. However, administrative data are also limited by underdiagnosis as well as potential disease misclassification based on International Classification of Diseases codes [1,13].

The general neglect of fungal diseases is likely to affect socially disadvantaged populations and reinforces existing health disparities. Fungal health disparities documented in the literature include Asian and Native Hawaiian or other Pacific Islander persons having blastomycosis incidence rates six times higher than White persons; American Indian or Alaska Native persons having coccidioidomycosis incidence rates four times higher than White persons; Black persons having the highest incidence rates of candidemia; and Black, American Indian or Alaska Native, or Hispanic persons having higher rates of all fungal deaths compared to other groups during the COVID-19 pandemic [2,3,4]. A deeper understanding of fungal disease health disparities related to age, race/ethnicity, sex, occupation, cultural, and sociodemographic factors and driven by social determinants of health could identify populations at greater risk so that effective strategies could be developed to reduce these disparities. For example, populations with inadequate access to nutritious food and high-quality healthcare might have a higher prevalence of uncontrolled diabetes, a risk factor for several severe fungal diseases [14]. Also, vulnerable populations may be overrepresented in highly endemic regions or in certain occupations due to long-standing structural violence, leading to an increased risk of environmental exposure. Policies and programs that address chronic occupational, housing, and healthcare access discrimination will likely play an important role in decreasing disparities in fungal diseases and other infectious diseases [2]. Increased data collection and the implementation of strategies to improve fungal disease diagnosis and treatment may also help decrease the burden of fungal diseases on disproportionately affected populations.

### 2.2. Diagnostic Tools

A prompt and accurate diagnosis is critical to the management of fungal infections. Many current diagnostic tools that are considered gold-standard, like cultures, have low sensitivities and a long turn-around time. Histopathology and microscopy methods usually provide rapid results but require highly qualified personnel, and sensitivity may be variable [15]. Other fungal assays like β-D glucan, *Aspergillus* galactomannan, and polymerase chain reaction (PCR) tests are available but are limited to referral centers, which increases turnaround times [15,16,17]. Developing and validating rapid, accurate, and cost-effective fungal diagnostic tests can improve both clinical and public health practice. An example is the lateral flow assay (LFA) test to detect the cryptococcal antigen (CrAg). This test provides a rapid result, is easy to perform, has a high sensitivity and specificity, and can detect the CrAg in serum before meningitis onset [18,19,20]. A molecular test for *Candida* in blood samples, the T2Candida assay, also shows promise for rapid detection [21]. However, few other molecular tests have been developed, and those that have are *in-house* tests or are not widely commercially available. Artificial intelligence (AI) may provide an innovative route to diagnose fungal infections by analyzing clinical images where diagnostic capacity is limited, particularly fungal diseases with skin and pulmonary manifestations; databases that inform AI models could be expanded, and further evaluation of this technology in clinical settings is needed [22].

Optimizing strategies for using fungal diagnostic tests in relevant clinical populations can help those most at risk for severe disease and death [23]. The identification of target populations and determination of the timing of certain diagnostic tests in relation to disease onset could be incorporated into optimal diagnostic algorithms. Establishing best practices for diagnostic stewardship, including standardized lab testing methods, could help improve patient outcomes. Fungal pneumonias are often misdiagnosed and patients are given several rounds of antibacterials; the development of a pan-pathogen standard respiratory diagnostic panel could also improve diagnostic stewardship [24,25,26]. Reducing the misdiagnosis of fungal infections and avoiding unnecessary antibacterial treatment and invasive diagnostic procedures can reduce the likelihood of the emergence of resistant fungal pathogens [17]. 

### 2.3. Therapeutics

The development of strategies for the early prevention, detection, and treatment of many fungal diseases can prevent severe, disseminated, and chronic infections. Fungal disease test and treat strategies can be implemented to prevent deaths, particularly from cryptococcosis, histoplasmosis, and talaromycosis in people living with advanced HIV, and should be expanded further. For example, in Guatemala, the rapid diagnosis and treatment of opportunistic fungal infections decreased the overall mortality by 7% over a year [23]. Antifungal agents, like the expanded spectrum triazoles, may have less severe side effects and a decreased risk of harmful drug–drug interactions compared to older agents like itraconazole. A lack of superiority clinical trials limits their use; further large, retrospective studies can evaluate their effectiveness compared with current recommended therapies. Pipeline antifungals with novel mechanisms of action (i.e., Gwt1 protein inhibitor, pyrimidine synthesis inhibitor, glucan synthase inhibitor) have shown promise through in vitro studies and in clinical trials and could address the difficulties in treating resistant fungi [27].

Antifungal resistance and tolerance are growing problems globally, both from treatment pressure and the overuse of fungicides that have similar chemical structures to human antifungals in the environment [28]. Research to further understand trends in antifungal resistance and to define resistance mechanisms can aid in the establishment of best treatment practices for resistant infections. Further work to develop rapid tests to detect resistance using molecular strategies is also needed. AsperGenius^®^, for example, is a multiplex real-time PCR test to detect azole resistance in *Aspergillus fumigatus* [29]. 

Antifungal prophylaxis can decrease morbidity and mortality in patients at risk of severe fungal infections; the assessment of targeted implementation strategies can improve the utilization of antifungal prophylaxis. Therapeutic drug monitoring (TDM) is recommended by fungal treatment guidelines for itraconazole, voriconazole, and posaconazole, particularly for those hospitalized and with severe disease; recent data show that less than 50% of prescribers use TDM for patients in whom it is indicated [30]. Investigation into barriers of TDM use and the overall understanding of appropriate TDM levels for antifungal agents can improve patient outcomes.

### 2.4. Vaccines

Vaccines have the potential to eliminate or reduce the burden of fungal infections. However, research efforts to develop fungal disease vaccines have been hampered by the eukaryotic nature of fungal pathogens and identifying vaccine targets that are specific to fungal organisms while avoiding cross-reactions with human cells [31]. Promising strides have been made in fungal disease vaccine development, including having three vaccines reach human clinical trials in the past (two recombinant *Candida* vaccines and one *Coccidioides* vaccine) [31]. A more recent coccidioidomycosis vaccine appears to be effective in canines, prompting calls for a human vaccine [32]. As researchers better understand the pathogenicity of certain fungal infections like *Candida*, novel vaccine targets could be identified and potentially provide cross-protection for several fungal pathogens [33]. Research should be pursued to develop effective vaccines and identify specific populations who would benefit most from immunization. 

**Table 1 jof-09-00820-t001:** Public Health Research Priorities that span fungal diseases.

Topical Area	Research Priorities
Burden and distribution of fungal diseases	Better characterize the public health, economic, health system, and personal burden imposed by fungal diseases Further understand disparities related to age, social determinants of health, race/ethnicity, sex, and other sociodemographic factors in order to identify and implement effective strategies to reduce disparities Implement global surveillance to track the fungal disease distribution worldwide and early warning systems to identify emerging fungal diseases Better understand associations between fungal infections and respiratory viral illnesses, such as COVID-19 and influenza Better define the geographic range and environmental niches of environmental disease-causing fungi Study prevention measures to reduce exposure to disease-causing fungi, such as early warning systems, personal protective equipment, and environmental mitigation strategies Investigate seasonal, weather, and climate-related patterns of fungal infections and evaluate the impact of climate change on established and emerging fungal diseases
Diagnostic tools	Develop and validate fungal diagnostic tests that are rapid, accurate, and cost-effective Develop rapid tests to identify antifungal-resistant fungal infections Optimize strategies for using fungal diagnostic tests in targeted clinical populations Establish best practices for fungal disease testing, including standardized testing Reduce the misdiagnosis of fungal infections and avoid unnecessary antibacterial treatment and invasive diagnostic procedures Develop strategies for the early detection and treatment of fungal diseases to prevent severe, disseminated, and chronic infections.
Therapeutics	Improve the understanding of the role of newer antifungal agents in disease management Further understand trends in antifungal resistance, including mechanisms through which resistance develops and best treatment practices for resistant infections Improve the understanding of appropriate populations for antifungal prophylaxis and optimal prophylaxis strategies Further develop “screen and treat” implementation strategies to prevent deaths in people living with advanced HIV Investigate barriers to therapeutic drug monitoring use and further research appropriate therapeutic drug monitoring levels
Vaccines	Develop effective vaccines and identify specific populations who would most benefit from immunization Further explore mRNA vaccine technology for vaccines against fungal diseases

## 3. Disease-Specific Priorities (Table 2)

### 3.1. Endemic Mycoses

Coccidioidomycosis, histoplasmosis, and blastomycosis are environmental fungal diseases with an expanding geographical range in the United States, with more than 21,000 cases reported in the United States each year, and probably globally as well [2]. Improved surveillance, in terms of the number of states reporting and data elements collected, could identify changes in the geographical distribution in addition to exposure sources, at-risk populations, clinical courses, and treatment patterns. 

The environmental reservoirs of fungi that cause endemic mycoses are not well defined, which limits the ability to understand how people are infected and develop effective prevention measures. Furthermore, a better understanding of the factors that determine the distribution of these fungi in the environment is needed for the development of accurate distribution maps and identification of the effects of climate change. Soil and climate parameters influence the distribution of *Coccidioides* species and could be explored further; the impact of rodents on *Coccidioides* species is not well understood and more research is needed [34,35]. The association with *Histoplasma* species and bat and bird manure is well established; however, the soil pH, land cover, and distance from water may impact the distribution, but more data are needed [36,37]. Although unanswered questions remain about the environmental habitats of all endemic fungi, this knowledge gap is especially severe for *Blastomyces* species [34,35,37,38,39]. Very little is known about where *Blastomyces* species occur in the environment and how people are exposed. This lack of information hampers outbreak investigations and efforts to develop *Blastomyces* distribution models. Technical challenges (i.e., complex sample types, difficulty of culture, BSL3 requirements) complicate the detection of endemic fungi in environmental samples; the lack of robust detection methods limits the ability to address the fungi distribution. Overall, more sensitive methods for the detection and monitoring of endemic fungi in the environment are also needed to understand risk and prevent human infections. Modeling techniques to examine potential geographical expansion could be enhanced to guide public health action. 

These fungal diseases often present as community-acquired pneumonia (CAP). Better quantifying the proportion of CAP and other lower respiratory infections caused by *Coccidioides*, *Histoplasma*, and *Blastomyces* could encourage the inclusion of these mycoses in CAP testing guidance and other public health interventions [40] Further assessments of the test sensitivity and specificity of coccidioidomycosis, histoplasmosis, and blastomycosis diagnostics, including inter-laboratory and inter-manufacturer assessment of consistency and performances, can improve testing practices. Next-generation sequencing, targeted fungal PCR assays, and other novel diagnostic approaches could prevent severe complications from delayed or missed diagnoses [41].

New treatments for coccidioidomycosis, histoplasmosis, and blastomycosis could be further explored, particularly with new triazole agents for histoplasmosis and blastomycosis. Currently, itraconazole is the mainstay therapeutic agent in mild-to-moderate and step-down therapy; however, it is fraught with side effects and drug–drug interactions [42,43]. Treatment failure in the endemic mycoses is also not well understood. The increased use of antifungal susceptibility testing could improve the understanding of the resistance rates over time, geographical distribution, and potential associations with environmental fungicide use. Clusters of resistance could be further investigated at a patient-level to identify risk factors for developing resistant isolates. 

Other endemic mycoses, like emergomycosis, paracoccidioidomycosis, and talaromycosis, occur more commonly in low- and middle-income countries. Risk factors for disease acquisition are not well understood; research in this area could inform prevention strategies. Investment in rapid, accurate non-invasive diagnostic tests that do not require referral laboratories is needed to better understand the incidence and prevent delayed diagnoses. An increase in antifungal susceptibility testing could guide treatment and prevent the use of ineffective antifungals. 

### 3.2. Invasive Candida Infections

Invasive *Candida* infections are a common cause of healthcare-associated infections, with an estimated 22,660 infections occurring annually in the United States, and they are associated with a high morbidity rate [44]. *C. albicans* has historically been the most common species causing candidemia; however, non-*albicans* species are being reported at an increasing rate globally [45]. In the United States, non-*albicans* species account for over 50% of reported cases and tend to have higher rates of resistance. Drug-resistant *C. parapsilosis* and *C. tropicalis* have been detected and reported more often globally [46,47]. Robust surveillance systems can monitor changes in *Candida* species, identify the emergence of new and rare species, detect clusters of infections, help characterize patient features and outcomes, and monitor changes and the emergence of patient risk factors (e.g., infection drug use), which would inform opportunities for the prevention of invasive candidiasis or resistance [48]. The Emerging Infections Program (EIP) surveillance system hosted by the CDC is an active, population-based approach that could be modeled for invasive *Candida* surveillance; however, EIP surveillance for *Candida* currently only looks at bloodstream infections and collects data from only ten states in the United States [49]. The World Health Organization’s Global Antimicrobial Resistance and Use Surveillance System (GLASS) is incorporating fungi, starting first with *Candida* bloodstream infections, and it will provide new insights into the global understanding of infections [50].

The gold standard for diagnosing invasive candidiasis is a positive blood culture; however, the sensitivity of blood cultures ranges from 21–71% [51]. To improve sensitivity and facilitate a quicker diagnosis, increasing the use of culture-independent diagnostic tests is critical. Beta-D-Glucan and procalcitonin may be viable options to improve the sensitivity to invasive candidiasis when combined with blood culture [51]. While the recent development of novel classes of antifungals equips providers with a wider array of options for *Candida* treatment, further efforts are needed in antifungal supply-chain management to provide effective therapy in resource-limited settings [27,52]. 

### 3.3. Candida Auris Infection

*Candida auris* is an emerging, often multidrug-resistant, transmissible yeast capable of causing invasive disease associated with high mortality. Since being first identified in the United States, the number of cases has increased at an alarming rate, with over 4000 cases reported in 2021 [53,54]. While still rare, the number of pan-resistant and echinocandin-resistant *C. auris* isolates reported in the United States and globally is increasing, but research is needed to understand outcomes and appropriate treatment, particularly for these pan-resistant isolates [55,56]. Transmission occurs mostly in healthcare settings, predominantly in high-acuity post-acute care facilities, so improving infection control practices is critical to prevent the spread of *C. auris* [57]. The COVID-19 pandemic may have also contributed to the spread of *C. auris* cases, and further investigation into associations, like with infection control measures, is warranted [58]. 

Better understanding *C. auris* transmission in healthcare settings can decrease morbidity and mortality among hospitalized patients. Infection control can be improved by investigating best practices for C. *auris* surveillance and colonization screening, including potential roles for admission testing and point prevalence surveys. Implementation studies can better determine the most effective ways to introduce and sustain infection control best-practice measures. The development of a point-of-care test for C. *auris* would allow screening with more rapid results and a quicker implementation of appropriate isolation precautions to minimize opportunities for transmission. This would be particularly beneficial in long-term care facilities where the risk of transmission is highest but there is often no on-site laboratory capacity [59]. A better understanding of best practices for prevention and control, including standardized methods for disinfectant testing and new methods for disinfection like no-touch devices, can limit spread within healthcare facilities [60]. The standardization of methods to evaluate products for *C. auris* decolonization or a reduction in the skin burden can also contribute to infection control in health facilities. Methods to evaluate therapies to manage colonization and a regulatory framework for the approval of these therapies currently do not exist; establishing methods and frameworks can improve the understanding of decolonization strategies. Colonization can contribute to continued transmission and an increased risk of infection, highlighting its importance for further research on decolonization methods [61]. Although several live animal and ex-vivo skin colonization models have been developed, these models are not standardized, making the comparison between methods and results obtained by different groups difficult [62,63,64,65] 

Recent studies identified changes in the human skin microbiome that facilitate colonization by *C. auris*, which suggest that normal skin flora may help to resist the colonization by C. *auris* [66,67]. Specifically, it has been shown that the skin of healthy individuals and those who are not colonized by *C. auris* is dominated by fungi from the genus *Malassezia*. However, people colonized by *C. auris* have a low prevalence of *Malassezia* species on their skin. A better understanding of the interplay between the normal skin microbiome, *Malassezia* species, the skin immune response, and *C. auris* may hold a key to understanding the mechanisms of *C. auris* colonization and possibly to the development of effective decolonization strategies. The development of a co-colonization model for *C. auris* and *Malassezia* species may help to better understand the interactions between these fungi on the skin. 

The environmental niches of *C. auris* and factors that influence its abundance in the environment and role as a human pathogen are not well understood. Community reservoirs could be described further; wastewater surveillance may be utilized in exploring these reservoirs and identifying new introductions in communities, although the implementation of wastewater surveillance needs to be further studied [68]. Addressing the broader evolutionary questions about the origins of *C. auris* and the reasons for the simultaneous emergence of four different clades may help to predict and possibly even control future emerging pathogens [69,70,71,72]. *C. auris* environmental isolates were found on a salt marsh and sandy beach in the Andaman Islands and in the sea off of Colombia; however, the association between human infections and the environment needs to be explored further [73,74]. Differences between the five primary clades of *C. auris* needs further research, particularly in relation to transmissibility and clinical manifestations. Whole-genome sequencing can be used to better recognize the transmission of the different clades. Additionally, novel clades may be emerging, and more robust surveillance may be able to identify these clades [75,76].

Treatment outcomes in patients with echinocandin-resistant or pan-resistant *C. auris* infections are unknown. Research into risk factors for developing these resistant strains is critical for controlling the spread of the yeast. Novel antifungals, like ibrexafungerp and fosmanogepix, appear promising in the treatment of *C. auris*; studies should continue to assess their efficacy and safety in different cohorts of infected people [77,78]. 

### 3.4. Aspergillus Fumigatus and Other Mold Infections

Invasive aspergillosis occurs primarily in immunocompromised people and causes at least 14,800 hospitalizations in the United States annually [13]. However, aspergillosis is not nationally notifiable in the United States, so the true burden is not well understood; expanded surveillance systems could better define its burden and distribution [59]. 

Antifungal-resistant *Aspergillus fumigatus* infections may be resulting from the overuse of agricultural fungicides [79,80,81]. Evaluating the impacts of agricultural, medicinal, and other environmental uses of antifungal compounds can provide insight into factors influencing this emergence and inform public health action. 

Healthcare-associated mold infections (e.g., aspergillosis, mucormycosis) are challenging to prevent and can cause substantial morbidity and mortality, particularly in immunocompromised people and newly emerging risk groups like those with COVID-19 or influenza [10,82]. Establishing best practices for preventing healthcare-associated mold infections, including environmental evaluations, ventilation system maintenance, healthcare linen safety standards, and spore concentration monitoring, is critical to save lives. Infection preventionists, for example, can develop and use checklists to evaluate the health facility management of bed linens [83].

### 3.5. Common, Non-Invasive Fungal Infections

Dermatophyte infections (over 4.9 million outpatient visits yearly) are common outpatient infections that can be misdiagnosed when visual inspection is the sole diagnostic tool [13]. Improved diagnosis and guideline-based treatment of fungal skin infections might improve patient outcomes and decrease resistance. Antifungal-drug-resistant tinea infections (e.g., those caused by *Trichophyton indotineae* and terbinafine-resistant *T. rubum*) are an emerging global problem [84,85], yet few providers in the United States routinely order diagnostic testing for tinea, limiting the ability to detect resistant infections [86]. Access to diagnostic testing for identification and susceptibility is poorly understood; a better understanding could help improve diagnostic stewardship. Tracking the emergence, particularly in limited-resource settings, can improve our understanding and allow targeted interventions to reduce the burden. 

Non-invasive *Candida* infections, like vulvovaginal candidiasis (VVC) and oral/esophageal candidiasis, cause over 3.6 million outpatient visits yearly in the United States [13]. VVC infections alone cause an estimated 1.4 million outpatient visits annually, and over 50% of women report having VVC at least once in their life [13,87]. Few providers routinely order diagnostic testing for VVC to distinguish between causes of vaginitis, and the high rates of empiric treatment suggest an overprescription of antifungals and misuse of antibacterial agents [88]. Examining best practices to improve clinical care can progress antifungal stewardship and patient outcomes. Innovative diagnostic methods, like point-of-care testing, can improve diagnostic stewardship as well. 

Recurrent VVC continues to increase globally and causes a substantial burden to patients and healthcare systems [89]. Studies that examine the risk of recurrent VCC with certain therapies, like Sodium-glucose-cotransport-2 (SGLT2) inhibitors or hormone replacement therapies, may identify interventions to mitigate their contribution to recurrent VVC [90,91,92]. Newer antifungal therapies developed to treat recurrent VCC appear promising; further studies can be conducted to look at their effectiveness in avoiding relapse [93,94]. The NDV-3A vaccine has shown promise in clinical trials to prevent recurrent VVC and further studies are ongoing [95]. Surveillance systems can help improve our understanding of the prevalence of and risk factors for non-invasive candidiasis. 

### 3.6. Neglected Tropical Fungal Diseases, including Mycetoma and Chromoblastomycosis

The World Health Organization names two fungal diseases as neglected tropical diseases (NTDs): mycetoma and chromoblastomycosis, as well as adding “other deep fungal infections” [96]. Their burden, epidemiology, and environmental reservoirs remain poorly documented as these NTDs are not nationally notifiable in many nations where they are known to be endemic. Establishing national surveillance systems within endemic regions could help identify at-risk populations, distribution patterns, and potential public health interventions to reduce disability. Identifying modes of transmission can inform control and prevention strategies. 

Mycetoma commonly occurs in resource-limited settings, away from medical centers with the capacity to perform advanced diagnostic tests to identify the causative organism. Clinically, mycetoma resembles other neglected skin diseases and can present either as bacterial (actinomycetoma) or fungal (eumycetoma) in origin. The development of accurate and cost-effective diagnostic modalities to diagnose mycetoma in field settings is critical to the treatment and further understanding of this fungal disease. Lateral flow antigen assays, a serologic diagnostic test to differentiate actinomycetoma from eumycetoma, or modifications to molecular techniques to make them more readily available at health outposts could transform mycetoma diagnostics [97]. Treatment for mycetoma is often long and unsuccessful; side effects from the frequently used itraconazole impact patient adherence [98]. Research into shorter, safer, more effective treatments and regimens is necessary to reduce the morbidity caused by mycetoma. The impact of accessible pharmacy services in rural settings to improve adherence and manage side effects could also be measured [98]. Stigma often accompanies mycetoma and has severe socioeconomic consequences; interventions to reduce stigma in populations affected by this fungal skin disease can be assessed and expanded if effective [99].

Chromoblastomycosis requires similar research priorities as mycetoma. Affordable, point-of-care testing to a species-specific level can help initiate and guide therapy quickly, as the average time from disease presentation to diagnosis is 9 years [100]. Shorter, more effective treatment regimens and the potential of topical immunotherapy could be studied to improve outcomes and adherence [101]. Certain host specific factors may lead to a more disfiguring disease and can be examined more closely. Assessments on chromoblastomycosis’s impact on quality of life could be completed to identify community-based interventions to improve the lives of those with this fungal disease. 

### 3.7. Sporotrichosis

Sporotrichosis causes approximately 2 cases per 1 million persons and 0.35 hospitalizations per 1 million persons annually in the United States [102,103]. Geographic niches and risk factors are not well understood; further knowledge on where these infections may occur more frequently can guide education to increase clinician awareness, which may improve diagnosing and treatment. Certain medical interventions, like glucocorticoids and TNF-α inhibitors, may increase the risk of acquiring sporotrichosis that leads to hospitalization [103]. Further research of the impact of these interventions on disease acquisition and severity is warranted and robust surveillance systems can monitor for increases in *Sporothrix* infections. 

Cat-transmitted sporotrichosis (CTS), caused by *Sporothrix brasiliensis*, is an emerging zoonotic fungal disease that has become a major public health concern in Brazil. *S. brasiliensis* transmission is maintained through cat-to-cat transmission, frequently occurring between feral or outdoor cats. Public health surveillance for CTS in humans and cats is limited. Published reports suggest that cases of CTS have rapidly increased in recent years, and cases are being reported in new geographic regions [104,105]. Sporotrichosis is frequently misdiagnosed; cases are frequently symptomatic for more than half a year before sporotrichosis diagnosis and treatment [106]. A promising LFA for CTS has been developed, which may decrease misdiagnoses; however, it is not widely available, and further assessment is needed [107].

Current disease control efforts and priorities utilize the One Health approach, targeting improving diagnostics through the education of medical and veterinary professionals, increasing laboratory capacity for *S. brasiliensis* testing, and estimating the disease burden in both human and cat populations. Future efforts for disease prevention should include the development of rapid diagnostic methods that reliably differentiate *S. brasiliensis* from other species of *Sporothrix*, along with new therapeutics or vaccinations targeted at preventing transmission among feral cats. The intervention should be cheap, effective, safe, long-lasting, and easy to integrate into existing infrastructure (e.g., veterinary diagnostic labs and veterinary clinics) and programs (e.g., sterilization campaigns, vaccination clinics). 

### 3.8. Pneumocystis Pneumonia

*Pneumocystis* pneumonia (PCP) is a potentially life-threatening infection caused by *Pneumocystis jirovecii.* PCP is believed to be one of the most common opportunistic infections in people living with HIV/AIDS in the United States and globally [108,109]. National surveillance for PCP in the United States does not currently exist; thus, the true burden is not known. In 2014, an estimated 10,590 hospitalizations in the United States involving PCP occurred [13]. A further understanding of the incidence can be obtained through expanded and standardized surveillance systems for immunocompromised and immunocompetent populations. More research into risk factors associated with PCP morbidity and mortality can increase the understanding of optimal prophylaxis strategies, particularly among non-HIV populations. *P. jirovecii* is a non-culturable fungus, histopathology has a low sensitivity, and diagnosis often relies on clinical symptoms and radiographic findings. Newer diagnostic tests are less invasive and more reliable; however, they have difficulties in distinguishing between infection and colonization. Investigation into cut-offs for colonization along with incorporating these newer diagnostic methods into resource-limited settings can improve diagnostic stewardship [110].

### 3.9. Opportunistic Fungal Infections in HIV

Certain fungal infections occur commonly as opportunistic infections in patients living with uncontrolled or advanced HIV infection. In addition to *Pneumocystis jirovecii*, described above, *Cryptococcus neoformans* (cryptococcal meningitis causes approximately 79,000–134,000 annual deaths globally), *Histoplasma capsulatum* (mortality rates can range from 10-60%), and *Talaromyces marneffei* (previously known as *Penicillium marneffei;* in some countries, it accounts for 4–11% of AIDS-related admissions) are known to cause substantial morbidity and mortality in these patients [111,112]. However, the disease prevalence among children living with HIV is not well understood [19]. Surveillance systems that can extract information on patient age can promote a better understanding of these fungal diseases. Point-of-care diagnostic tests have been developed for *Cryptococcus neoformans;* similar, affordable, field-ready diagnostic tools may be prioritized for these other fungal infections to prevent delays in diagnosis and treatment [19]. A recently developed *Histoplasma* antigen detection lateral flow assay may become the standard diagnostic tool for disseminated histoplasmosis, but further evaluation is needed among hospitalized and outpatient settings [113]. Current and future point-of-care diagnostic tests could expand their use with data from non-immunocompromised individuals. Co-infections, like tuberculosis, commonly occur in patients living with HIV who have opportunistic fungal infections [19,113]. Concurrent treatment for both diseases could be investigated to look at patient outcomes, tolerability, and safety in different medical settings.

**Table 2 jof-09-00820-t002:** Public health research priorities for specific fungal diseases.

Fungal Disease	Research Priorities
Endemic mycoses (e.g., coccidioidomycosis, histoplasmosis, and blastomycosis)	Better quantify the proportion of community-acquired pneumonia (CAP) and other lower respiratory infections caused by endemic mycoses Identify risk factors and determinants for pulmonary and disseminated disease Further understanding of climate change’s impact on their geographic distribution Consideration of a cross-pathogen “failure to respond” to initial CAP antibiotics guideline Further assessments of diagnostic test sensitivity and specificity, including inter-lab and inter-manufacturer assessment of consistency
Invasive *Candida* infection	Detect changes in the prevalence of *Candida* species and identify the emergence of new *Candida* species and other yeasts and resistance Identify and study measures to prevent invasive candidiasis Monitor changes in risk factors and identify new risk factors (e.g., injection drug use) for invasive candidiasis Understand the role of culture-independent diagnostics tests in diagnosing invasive candidiasis Describe the prevalence of different types of invasive candidiasis and outcomes (i.e., mortality, secondary complications)
*Candida auris* infection	Better understand of *C. auris* colonization and transmission in healthcare settings Investigate best practices for *C. auris* surveillance and colonization screening Ensure best practices for *C. auris* infection prevention and control, including standardized methods for disinfectant testing and screening Investigate methods to decolonize or reduce the skin burden of *C. auris*, including standardized methods to evaluate products Investigate the environmental niches of *C. auris* and its impact on human disease Better understand community reservoirs of *C. auris* and the utilization of wastewater surveillance Better understand clinical and phenotypic differences between *C. auris* clades Understand the role of community transmission in the spread of *C. auris*
*Aspergillus fumigatus* and other mold infections	Evaluate the impacts of agricultural, medicinal, and other environmental uses of antifungal compounds on the selection for azole resistance Establish best practices for preventing healthcare-associated mold infections (e.g., aspergillosis, mucormycosis), including environmental evaluations, ventilation system maintenance, healthcare linen safety standards, and spore concentration monitoring
Common, non-invasive fungal infections (e.g., vulvovaginal candidiasis, dermatophyte infections)	Promote improved diagnosis and guideline-based treatment of superficial fungal infections, such as vulvovaginal candidiasis and dermatophyte (ringworm) infections Track the emergence and reduce the burden of severe or antifungal-resistant dermatophyte infections, particularly in limited-resource settings Identify ways to improve diagnostic testing practices for vulvovaginal candidiasis Better understand the prevalence of and risk factors for non-invasive candidiasis (i.e., oral, esophageal, vaginal)
Neglected fungal tropical diseases (e.g., mycetoma, chromoblastomycosis)	Understand the burden, epidemiology, and environmental reservoirs of neglected tropical fungal diseases Develop accurate and cost-effective methods to diagnose neglected tropical fungal diseases Reduce disability from neglected tropical fungal diseases
Sporotrichosis	Improve surveillance to identify high-prevalence areas of sporotrichosis for provider education and awareness Assess the impact of medical interventions on disease acquisition and severity Improve diagnostics through education of medical and veterinary professionals and increase laboratory capacity for *S. brasiliensis* testing Estimate the disease burden in both human and cat populations Develop new therapeutics or vaccinations targeted at preventing transmission among feral cats
*Pneumocystis* pneumonia	Further understanding of the incidence Identification of risk factors associated with *Pneumocystis* pneumonia morbidity and mortality, particularly among non-HIV populations Investigation into colonization cut-offs for newer diagnostic methods
Opportunistic fungal infections in people living with advanced HIV (e.g., cryptococcosis, histoplasmosis, talaromycosis)	Better understand prevalence of opportunistic fungal infections in children living with HIV Improve diagnostic testing through the development of affordable, point-of-care diagnostic tools Investigate outcomes and adverse effects of concurrent therapy with other non-fungal HIV co-infections like tuberculosis

## 4. Conclusions

Many aspects of fungal disease distribution, diagnosis, treatment, and prevention are not well understood. Diagnostic testing is critical as appropriate treatment, the understanding of the burden and distribution, and prevention efforts hinge on being able to correctly identify fungal pathogens. A multidisciplinary approach to researching fungal pathogens can improve understanding and strengthen partnerships within the fungal research community. This research agenda can provide a starting point to identify current knowledge gaps for action. Research that does not reach the front-line healthcare cadre will have a limited impact, and diagnostic tools developed but not accompanied by effective communication may not be used. It is vital that research is partnered with appropriate communication and dissemination to healthcare providers, public health professionals, and policy makers to help raise awareness of fungal diseases. 

## Data Availability

Not applicable.

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
