# Peer review of "Public Health Research Priorities for Fungal Diseases: A Multidisciplinary Approach to Save Lives"

_jof, 2023, doi:10.3390/jof9080820_

Round 1
Reviewer 1 Report
The authors aim to outline overarching and disease-specific public health research priorities for fungal diseases, against the background of the recently published WHO FPPL.
The manuscript is well written. Section 2 is nevertheless very dense with in my opinion to many generic statements without being specific and/or providing some illustrative examples.
I have the following comments for the authors:
The abstract is missing what is the aim of the manuscript as well as what can the reader except in the manuscript, and/or what the scope is of the manuscript. It reads too much as a number of statements, and not a coherent concise summary of the manuscript.
Line 48, ‘Most fungal diseases are not reportable to’, could the authors state which ones are reportable?
Line 53, wondering if there are any examples of surveillance systems for fungal diseases and/or antifungal susceptibility/resistance which could be mentioned here?
Line 86, ‘Histopathology and microscopy methods usually have a rapid response’, not sure what is meant here?
Line 88, include galactomannan here
Line 97-99, could the authors provide an example where AI has been shown useful in diagnosing skin abnormalities or pulmonary abnormalies on imaging?
Line 113, provide a successful example of a ‘test and treat’ strategy
Line 128/129, provide an example (e.g. Aspergenius to detect azole resistance in Aspergillus fumigatus)
Line 132, make this a bit more specific, e.g. this relates mainly to the use of mould active azoles and provide an example
Line 289 and 290, the authors should include references to large observational studies in which outcomes of C. auris infections are reported
Line 300, work from Jo Rhodes and Matt Fisher have shown evidence of agricultural fungicides being responsible for infections in humans with azole-R A. fumigatus, the authors should refer to this paper.
Line 306, it is the question if patient with COVID associated pulmonary aspergillosis acquire their aspergillus in the hospital setting, and therefore they measures listed might not be sufficient; in addition the treatment administered for sever COVID pneumonia also plays a role here
Line 315, the authors should add a bit more specific data, e.g. which species, it is with respect to terbinafine resistant, started in India, is becoming global problem
Line 366, provide a reference to the use of immunotherapy
Line 409, Pneumocystis is not difficult to culture, it is a non-culturable fungus
Section 3.8 is an odd one in the list (all the other one are specific fungi/fungal diseases and this one is referring to an at risk group), I would recommend to dedicate this section to Cryptococcus.
is of acceptable quality
Author Response
Reviewer 1:
The abstract is missing what is the aim of the manuscript as well as what can the reader except in the manuscript, and/or what the scope is of the manuscript. It reads too much as a number of statements, and not a coherent concise summary of the manuscript.
- Added the scope of the manuscript into the abstract and created better flow.
Line 48, ‘Most fungal diseases are not reportable to’, could the authors state which ones are reportable?
- Added examples of reportable fungal diseases and referenced CDC website on which fungal diseases are reportable
Line 53, wondering if there are any examples of surveillance systems for fungal diseases and/or antifungal susceptibility/resistance which could be mentioned here?
- Added a line about the National Notifiable Disease Surveillance System.
Line 86, ‘Histopathology and microscopy methods usually have a rapid response’, not sure what is meant here?
- Changed to rapid results
Line 88, include galactomannan here
- Included galactomannan
Line 97-99, could the authors provide an example where AI has been shown useful in diagnosing skin abnormalities or pulmonary abnormalies on imaging?
- The authors have cited a recent study where researchers used AI to aid in diagnosis of mycetoma
Line 113, provide a successful example of a ‘test and treat’ strategy
- Added a successful example from Guatemala.
Line 128/129, provide an example (e.g. Aspergenius to detect azole resistance in Aspergillus fumigatus)
- Added AsperGenius and citation to the text.
Line 132, make this a bit more specific, e.g. this relates mainly to the use of mould active azoles and provide an example
- Added the specific antifungals that TDM is recommended for.
Line 289 and 290, the authors should include references to large observational studies in which outcomes of C. auris infections are reported
- Changed this sentence to just pan-resistant or echinocandin resistant auris in which outcome data are limited.
Line 300, work from Jo Rhodes and Matt Fisher have shown evidence of agricultural fungicides being responsible for infections in humans with azole-R A. fumigatus, the authors should refer to this paper.
- Referenced their paper in this sentence
Line 306, it is the question if patient with COVID associated pulmonary aspergillosis acquire their aspergillus in the hospital setting, and therefore they measures listed might not be sufficient; in addition the treatment administered for sever COVID pneumonia also plays a role here
- Added a line before this sentence about prevention is challenging to allude to the difficulty in preventing infections
Line 315, the authors should add a bit more specific data, e.g. which species, it is with respect to terbinafine resistant, started in India, is becoming global problem
- Added the two specific species here.
Line 366, provide a reference to the use of immunotherapy
- Provided a reference
Line 409, Pneumocystis is not difficult to culture, it is a non-culturable fungus
- Changed to non-culturable fungus
Section 3.8 is an odd one in the list (all the other one are specific fungi/fungal diseases and this one is referring to an at risk group), I would recommend to dedicate this section to Cryptococcus.
- We think it is important to highlight this at-risk group since they have poor outcomes and funding is dedicated to these populations.
Reviewer 2 Report
The review from Smith, et al., is a nicely written reminder that fungal diseases cause significant morbidity and are largely understudied. The authors discuss major priorities like development of diagnostic tools, better therapeutics, and vaccines. They then discuss disease-specific priorities. This was informative, but the priorities for individual fungi are largely repetitive because of the overall problems described in the section 2. However, this does not limit the importance of the paper. I have only a few minor comments.
1. It might be useful to remind readers that Talaromyses marneffei was previously named Penicillium marneffei. Keeping track of naming conventions in fungi can be daunting.
2. There are a few sentences with editing errors like in line 211 that reads “…are been detected…” and lines 234-235 that reads “… with over 4,000 cases were reported in 2021”. The authors might want to get an editor to reread the manuscript for these types of things.
3. Line 409 says that P. jirovecci “…..often proves difficult to culture” but in reality there is no long term culture system that has been reproduced by the field. Table 2 indicates that a research priority might be “Further understanding of the geographic distribution and incidence” of Pneumocystis pneumonia. This pathogen has been shown by the Kovacs lab to be devoid of genes that would allow it to live in the environment so there is likely no specific geographical range outside of where people live. This idea wasn’t discussed in the section on P. jirovecii in the text and maybe should be developed if put on Table 2.
English is very good. Just a few editing errors or typos detected.
Author Response
Reviewer 2:
- It might be useful to remind readers that Talaromyses marneffei was previously named Penicillium marneffei. Keeping track of naming conventions in fungi can be daunting.
- Added the older species name
- There are a few sentences with editing errors like in line 211 that reads “…are been detected…” and lines 234-235 that reads “… with over 4,000 cases were reported in 2021”. The authors might want to get an editor to reread the manuscript for these types of things.
- Updated these grammatical errors
- Line 409 says that P. jirovecci “…..often proves difficult to culture” but in reality there is no long term culture system that has been reproduced by the field. Table 2 indicates that a research priority might be “Further understanding of the geographic distribution and incidence” of Pneumocystis pneumonia. This pathogen has been shown by the Kovacs lab to be devoid of genes that would allow it to live in the environment so there is likely no specific geographical range outside of where people live. This idea wasn’t discussed in the section on P. jirovecii in the text and maybe should be developed if put on Table 2.
- Changed to non-culturable fungus in text and removed geographic distribution from the table.
Reviewer 3 Report
From the Public Health viewpoint, and in connection with the WHO's document on the list of priority fungal pathogens, a review is presented that includes the main worldwide fungal infections. The following is an overview of the priority factors for their study/improvement from the public health perspective: Burden and distribution of fungal diseases, diagnostic tools, therapeutics, and vaccines. Subsequently, a more detailed review of the available resources for managing different fungal infections is conducted, emphasizing the specific priorities for each case. While many of these priorities are already in progress, it is always helpful to be reminded of potential weaknesses.
The document is well-structured and written, with an abundant and updated bibliography. However, there are some aspects that should be revised or clarified.
1. In line 97, there is a reference to AI, but given the current debate surrounding this technology, it should be specified that it is an aid for analyzing images (X-ray, skin lesions, etc.).
2. The recent emergence of Candida auris in clinical settings is undoubtedly a threat due to its characteristic resistance to antifungal compounds and commonly used disinfectants. The authors present this fungus in a separate section (3.3), even though it is a particular case of invasive candidiasis (section 3.2).
Moreover, it's striking that out of a total of 105 references in the review, this microorganism accounts for 26 references. While I understand that some authors are involved in this topic, I believe this section is overrepresented compared to the others. Either this section is too extensive, or the others lack as much detail.
For example:
L243-263: The paragraph regarding transmission and colonization could be summarized more concisely.
L. 274-288: The explanation of the clades of C. auris is confusing. Please simplify.
However, in lines 227-229 there is a mention of new classes of antifungals, but no examples or bibliographic references are provided.
3. Regarding recurrent VVC, I miss a mention of the development of the vaccine NDV-3A (NovaDigm) currently in phase 3 clinical trials.
4. Opportunistic infections by Pneumocystis are usually associated with HIV patients. However, the increase in the number of patients with other immunodeficiencies poses a significant risk among non-HIV patients, where the fungal burden is lower and delayed diagnosis is associated with higher mortality rates than in HIV patients. In my opinion, this data in non-HIV patients further reinforces the need to implement surveillance and diagnostic measures in these immunocompromised patients.
5. In the section on other opportunistic infections in HIV patients, the mortality data for cases of Cryptococcus neoformans, Histoplasma capsulatum, and Talaromyces marneffei are difficult to compare (numbers, percentages, percentages in HIV patients). If already known, it would also be helpful to mention the preferred geographical distribution of these infections.
Minor errors in spelling/grammar:
L.163: "identification."
L. 214-215: "...prevention OF invasive candidiasis..." or "... invasive candidiasis prevention..."?
L. 281-283: "C. auris environmental isolates were found on a salt marsh and sandy beach in the Andaman Islands and in the sea off of Colombia; however, the association between human infections needs to be explored further." I don't understand the second part of this sentence. Is something missing?
Line 333: please, indicate the meaning of SGLT2.
Table 2. Regarding the priorities for C. auris infection:
"Better understanding of C. auris colonization and transmission in healthcare settings." Remove the second "of C. auris ."
"Investigate the environmental niches of C. auris and it’s impact on human disease." Do you mean ”its”?
The document is well-structured and written, with an abundant and updated bibliography.
I only appreciated some minor errors in spelling/grammar:
L.163: "identification."
L. 214-215: "...prevention OF invasive candidiasis..." or "... invasive candidiasis prevention..."?
L. 281-283: "C. auris environmental isolates were found on a salt marsh and sandy beach in the Andaman Islands and in the sea off of Colombia; however, the association between human infections needs to be explored further." I don't understand the second part of this sentence. Is something missing?
Line 333: please, indicate the meaning of SGLT2.
Table 2. Regarding the priorities for C. auris infection:
"Better understanding of C. auris colonization and transmission in healthcare settings." Remove the second "of C. auris ."
"Investigate the environmental niches of C. auris and it’s impact on human disease." Do you mean ”its”?
Author Response
Reviewer 3:
From the Public Health viewpoint, and in connection with the WHO's document on the list of priority fungal pathogens, a review is presented that includes the main worldwide fungal infections. The following is an overview of the priority factors for their study/improvement from the public health perspective: Burden and distribution of fungal diseases, diagnostic tools, therapeutics, and vaccines. Subsequently, a more detailed review of the available resources for managing different fungal infections is conducted, emphasizing the specific priorities for each case. While many of these priorities are already in progress, it is always helpful to be reminded of potential weaknesses.
The document is well-structured and written, with an abundant and updated bibliography. However, there are some aspects that should be revised or clarified.
- In line 97, there is a reference to AI, but given the current debate surrounding this technology, it should be specified that it is an aid for analyzing images (X-ray, skin lesions, etc.).
- Added “by analyzing clinical images” to the text
- The recent emergence of Candida auris in clinical settings is undoubtedly a threat due to its characteristic resistance to antifungal compounds and commonly used disinfectants. The authors present this fungus in a separate section (3.3), even though it is a particular case of invasive candidiasis (section 3.2).
Moreover, it's striking that out of a total of 105 references in the review, this microorganism accounts for 26 references. While I understand that some authors are involved in this topic, I believe this section is overrepresented compared to the others. Either this section is too extensive, or the others lack as much detail.
For example:
L243-263: The paragraph regarding transmission and colonization could be summarized more concisely.
- As C. auris is uniquely transmitted compared to other fungi, we believe this paragraph is important as research is needed to better understand and prevention transmission and colonization.
- 274-288: The explanation of the clades of C. auris is confusing. Please simplify.
- Simplified the explanation of the C. auris clades.
However, in lines 227-229 there is a mention of new classes of antifungals, but no examples or bibliographic references are provided.
- Added references to this sentence on novel antifungals.
- Regarding recurrent VVC, I miss a mention of the development of the vaccine NDV-3A (NovaDigm) currently in phase 3 clinical trials.
- Added on sentence on the NDV-3A vaccine for RVVC
- Opportunistic infections by Pneumocystis are usually associated with HIV patients. However, the increase in the number of patients with other immunodeficiencies poses a significant risk among non-HIV patients, where the fungal burden is lower and delayed diagnosis is associated with higher mortality rates than in HIV patients. In my opinion, this data in non-HIV patients further reinforces the need to implement surveillance and diagnostic measures in these immunocompromised patients.
- Changed the wording to focus on all immunocompromised patients
- In the section on other opportunistic infections in HIV patients, the mortality data for cases of Cryptococcus neoformans, Histoplasma capsulatum, and Talaromyces marneffei are difficult to compare (numbers, percentages, percentages in HIV patients). If already known, it would also be helpful to mention the preferred geographical distribution of these infections.
- Global data are not available robustly for the preferred geographical distribution. In most places, diagnostic testing is limited which makes it difficult to measure incidence. We pulled together the information known on these diseases, although, we realize they might not be comparable.
Minor errors in spelling/grammar:
L.163: "identification."
- Updated
- 214-215: "...prevention OF invasive candidiasis..." or "... invasive candidiasis prevention..."?
- Corrected
- 281-283: "C. auris environmental isolates were found on a salt marsh and sandy beach in the Andaman Islands and in the sea off of Colombia; however, the association between human infections needs to be explored further." I don't understand the second part of this sentence. Is something missing?
- Corrected
Line 333: please, indicate the meaning of SGLT2.
- Spelled out SGLT2 in text
Table 2. Regarding the priorities for C. auris infection:
"Better understanding of C. auris colonization and transmission in healthcare settings." Remove the second "of C. auris ."
- Updated
"Investigate the environmental niches of C. auris and it’s impact on human disease." Do you mean ”its”?
- Updated